# Micronutrient Status of Critically Ill Patients with COVID-19 Pneumonia

**DOI:** 10.3390/nu16030385

**Published:** 2024-01-29

**Authors:** Sander Rozemeijer, Henrike M. Hamer, Annemieke C. Heijboer, Robert de Jonge, Connie R. Jimenez, Nicole P. Juffermans, Romein W. G. Dujardin, Armand R. J. Girbes, Angélique M. E. de Man

**Affiliations:** 1Department of Intensive Care Medicine, Research VUmc Intensive Care (REVIVE), Amsterdam Cardiovascular Science (ACS), Amsterdam Infection and Immunity Institute (AI&II), Amsterdam Medical Data Science (AMDS), Amsterdam UMC, Location VUmc, Vrije Universiteit Amsterdam, De Boelelaan 1117, 1081 HV Amsterdam, The Netherlands; a.girbes@planet.nl (A.R.J.G.); ame.deman@amsterdamumc.nl (A.M.E.d.M.); 2Department of Anesthesiology, Amsterdam UMC, Location VUmc, Vrije Universiteit Amsterdam, De Boelelaan 1117, 1081 HV Amsterdam, The Netherlands; r.w.dujardin@amsterdamumc.nl; 3Department of Laboratory Medicine, Laboratory Specialized Techniques and Research, Amsterdam Gastroenterology Endocrinology and Metabolism, 1105 AZ Amsterdam, The Netherlands; h.hamer@amsterdamumc.nl; 4Department of Laboratory Medicine, Endocrine Laboratory, Amsterdam Gastroenterology Endocrinology and Metabolism, 1105 AZ Amsterdam, The Netherlands; a.heijboer@amsterdamumc.nl; 5Department of Laboratory Medicine, Amsterdam Gastroenterology Endocrinology and Metabolism, 1105 AZ Amsterdam, The Netherlands; r.dejonge1@amsterdamumc.nl; 6OncoProteomics Laboratory, Department Laboratory Medical Oncology, Amsterdam UMC, 1081 HV Amsterdam, The Netherlands; c.jimenez@amsterdamumc.nl; 7Department of Intensive Care and Laboratory of Translational Intensive Care, Erasmus Medical Center, 3015 GD Rotterdam, The Netherlands; n.p.juffermans@amsterdamumc.nl

**Keywords:** critically ill, COVID-19, micronutrients, malnutrition, deficiencies, blood levels, micronutrient administration, inflammation

## Abstract

Micronutrient deficiencies can develop in critically ill patients, arising from factors such as decreased intake, increased losses, drug interactions, and hypermetabolism. These deficiencies may compromise important immune functions, with potential implications for patient outcomes. Alternatively, micronutrient blood levels may become low due to inflammation-driven redistribution rather than consumption. This explorative pilot study investigates blood micronutrient concentrations during the first three weeks of ICU stay in critically ill COVID-19 patients and evaluates the impact of additional micronutrient administration. Moreover, associations between inflammation, disease severity, and micronutrient status were explored. We measured weekly concentrations of vitamins A, B6, D, and E; iron; zinc; copper; selenium; and CRP as a marker of inflammation state and the SOFA score indicating disease severity in 20 critically ill COVID-19 patients during three weeks of ICU stay. Half of the patients received additional (intravenous) micronutrient administration. Data were analyzed with linear mixed models and Pearson’s correlation coefficient. High deficiency rates of vitamins A, B6, and D; zinc; and selenium (50–100%) were found at ICU admission, along with low iron status. After three weeks, vitamins B6 and D deficiencies persisted, and iron status remained low. Plasma levels of vitamins A and E, zinc, and selenium improved. No significant differences in micronutrient levels were found between patient groups. Negative correlations were identified between the CRP level and levels of vitamins A and E, iron, transferrin, zinc, and selenium. SOFA scores negatively correlated with vitamin D and selenium levels. Our findings reveal high micronutrient deficiency rates at ICU admission. Additional micronutrient administration did not enhance levels or expedite their increase. Spontaneous increases in vitamins A and E, zinc, and selenium levels were associated with inflammation resolution, suggesting that observed low levels may be attributed, at least in part, to redistribution rather than true deficiencies.

## 1. Background

Micronutrients play a key role in the innate and adaptive immunity of the human body [1,2,3]. In severe illnesses, such as hypoxemic COVID-19 disease, metabolic demands are high. In critically ill patients, micronutrient levels can become low due to decreased intake, increased losses, drug interactions with (single) micronutrients, and hypermetabolism. As a result, deficiencies can develop, and important immune functions might become impaired [4]. COVID-19 patients are especially at risk for deficiencies when they develop acute respiratory distress syndrome (ARDS), a katabolic state, and acute kidney injury necessitating renal replacement therapy, the risk of which may increase during prolonged ICU occupancy. High deficiency rates of vitamins A (71.7%), B6 (42.5%), and D (74.3%) have been described in the literature [5]. High prevalences (50–83%) of vitamin D deficiency are frequently reported in multiple COVID-19 studies [6,7,8,9,10,11,12]. Multiple small cohort studies in COVID-19 ICU patients also described low iron and transferrin levels at admission [13,14,15,16]. Low zinc and selenium levels have also been found [7,17,18,19].

Alternatively, low micronutrient levels in blood may be due to inflammation-driven redistribution rather than consumption [20]. These decreased plasma concentrations can spontaneously normalize after the inflammation is over and do not necessarily warrant supplementation. Therefore, establishing the exact micronutrient requirements in the critically ill is notoriously difficult. Covering the daily required intake of micronutrients for healthy persons is probably not sufficient [4], and readily available monitoring of blood concentrations is lacking because diagnostic tests are not rapidly available. This has resulted in a wide variation in pragmatic protocols for micronutrient administration in critically ill patients [21].

In this study, we investigated blood micronutrient concentrations in critically ill patients with hypoxemic COVID-19 during their first three weeks of ICU stay. These patients were infected with SARS-CoV-2 during the first wave and were quite homogenous regarding their hyper-inflammatory response. Up to now, the concentrations of different micronutrients have never been measured during three weeks of ICU stay. In addition, we explored the plasma micronutrient concentrations between two hospitals with a different micronutrient administration regime. Secondary aims were to investigate the association between inflammation, severity of disease, and micronutrient status.

## 2. Methods

### 2.1. Study Population and Design

Twenty critically ill ICU patients with COVID-19 pneumonia were included in this prospective biobank cohort study in Amsterdam UMC. Ten patients were admitted to the ICU of Amsterdam University Medical Centers, location AMC, Amsterdam, the Netherlands (hereafter AMC), and ten patients were admitted to the ICU of Amsterdam University Medical Centers, location VUmc, Amsterdam, the Netherlands (hereafter VUmc). Five trauma patients, without COVID-19, were included in AMC as controls. These patients were admitted to the Emergency Department (ED) with trauma. Thereafter, three were admitted to the ICU and two to the ward. In accordance with each hospital’s protocol, ICU patients in AMC received daily micronutrient administration by enteral nutrition with Peptamen^®^ Intense, and ICU patients in VUmc received daily micronutrient administration by enteral nutrition with Peptamen^®^ Intense, Nutrison^®^ Advanced Protison, Nutrison^®^ Protein Intense, Nutrison^®^ Protein Plus, Peptamen^®^ HN, or Fresubin^®^ 2 KCAL HP. In addition, these patients in VUmc intravenously received additional multivitamins (Cernevit^®^) and multitrace elements (Supliven^®^) once a day during 5 days in the first week of ICU stay. This resulted in an additional intravenous 17,500 IU vitamin A, 22.65 mg vitamin B6, 1100 IU vitamin D (D3), 51 mg vitamin E, 27 mg ferric chloride, 52.5 mg zinc, 865 µg selenite, and 5 mg copper during these 5 days. In the days thereafter, they orally received additional micronutrients (Supradyn^®^) once a day containing 800 µg (2667 IU) vitamin A, 2 mg vitamin B6, 5 µg (200 IU) vitamin D, 12 mg vitamin E, 14 mg iron, 10 mg zinc, 50 µg selenium, and 1 mg copper. See Appendix A for a specification of the ingredients of the standard enteral nutrition and the additional micronutrients. See Appendix A for an overview of the (enteral) nutrition each patient received in each hospital. In both hospitals, a few patients also received Smofkabiven^®^ (total parenteral nutrition, including vitamins and trace elements). In VUmc, two patients received the additional intravenous micronutrients again in week 2/3, possibly due to CVVH. Medical doctors, registered dietitians, and nutritional assistants were all involved in the multidisciplinary treatment of the included patients.

Blood samples and clinical data were obtained from the COVID-19 biobank of Amsterdam UMC, registered under number 2021.0016 (2020.182). The researchers complied with the “Open Science” policy, research code, and insurance policy of the biobank of Amsterdam UMC. Informed consent was obtained as an opt-out from all participants or their legal representatives.

### 2.2. Samples and Measurements

Heparin plasma and serum samples were collected from the included patients at the day of ICU admission and weeks 1, 2, and 3 of ICU stay. Samples were frozen on the day of collection and stored at −80 °C until analysis. In AMC, one patient was transferred to the ward at day 13, and in VUmc, one patient was transferred to the ward at day 18. In these two cases, blood samples were taken on the ward after ICU discharge. The following micronutrient levels were determined at each time point: plasma vitamin A, whole-blood vitamin B6, plasma 25-hydroxyvitamin D (total 25OHD: sum of 25OHD2 and 25OHD3), 24.25diOHD3, vitamin D-binding protein (DBP), plasma vitamin E, plasma iron, serum zinc, serum selenium, and serum copper. Vitamin C could not be determined because none of the stored samples were acidified. C reactive protein (CRP), measured on the same day as the micronutrient, was used as an estimation of inflammation. For the control patients, the same micronutrients were determined at the day of admission at the ED. No follow-up samples were available because of their much shorter hospital stay compared with the COVID-19 patients.

The measurements were performed by the ISO15189:2012 accredited department of Laboratory Medicine using the routine Roche Cobas 8000 platform (Roche Diagnostics GmbH, Mannheim, Germany) for measurements of albumin, CRP, iron, ferritin, and transferrin according to the manufacturer’s instructions. DBP was measured using the enzyme-linked immunosorbent assay (ELISA) assay from Immundiagnostik AG (Bensheim, Germany). Chromatography coupled to mass spectrometry (LC-MS/MS) was used for measurements of 25OHD2, 25OHD3, and 24.25diOHD3, as previously published [22]. Vitamins A and E were measured in heparin plasma after extraction in hexane using HPLC [23]. Vitamin B6 in heparin whole blood was measured using a reverse-phase HPLC method with pre-column derivatization using semicarbazide [24]. Zinc, selenium, and copper were measured in serum samples using ICP-MS (Nexion 300X, Perkin Elmer, Waltham, MA, USA).

### 2.3. Power Analysis

This is an explorative study, and the number of samples is a convenience sample.

### 2.4. Statistical Analysis

We analyzed the data using IBM SPSS Statistics version 26. Variables were reported as mean ± standard deviation (SD) if normally distributed and as median [25th to 75th] percentile if not normally distributed. Normality was assessed using skewness results, histograms, and the Shapiro–Wilk test.

The difference in micronutrient levels between and within the two hospitals over time was tested with a linear mixed model analysis. Random effects were the 20 COVID-19 patients, and fixed effects were time, hospital, and the interaction between time and hospital. A possible significant difference in the increase or decrease over time between the hospitals was only evaluated between baseline and week 3. A *p*-value of <0.05 was considered to be statistically significant.

The prevalences of deficiency was separately calculated for each hospital, see Appendix A. These rates were not compared between the hospitals due to the low sample size. 

A Pearson product-moment correlation coefficient was computed to assess the relationship between CRP and SOFA score on the one hand and a micronutrient concentration on the other hand.

## 3. Results

In total, 20 COVID-19 patients and 5 control patients were included. A total of 10 critically ill patients were admitted to the ICU of AMC and 10 critically ill patients to the ICU of VUmc. The other 5 included patients were trauma patients admitted to the ED of AMC. The baseline characteristics are shown in Table 1.

At baseline, both critically ill COVID-19 patient groups were comparable regarding age, sex, BMI, medical history, and hospital admission scores. The control patients consisted of trauma patients, who differed, e.g. regarding age, compared with the critically ill COVID-19 patients (Table 1).

### 3.1. Vitamins

Vitamins A, B6, D, and E are shown in Figure 1. Vitamin A levels were significantly lower in the COVID-19 patients compared with the control at baseline (*p* < 0.001), with 100% of the patients being deficient in both hospitals and 0% in the controls. During 3 weeks of ICU stay, vitamin A levels increased, and the percentage of patients with deficiency decreased to 20–30% in both hospitals. Vitamin A levels only differed between the hospitals at week 1, being significantly higher in AMC patients (*p* = 0.047).

Vitamin B6 levels were not different between the COVID-19 and control patients (*n* = 3) at baseline, with 80 and 100% of the patients being deficient in AMC and VUmc, respectively, and 60% in the controls. Deficiency levels stayed >80% during 3 weeks of ICU stay. Vitamin B6 levels did not differ between the patients in both hospitals at any time.

Total 25OHD levels were decreased in the majority of patients, both with the COVID-19 (70% deficiency in both hospitals) and control patients (60% deficiency) at baseline. 25OHD levels increased a little during the time, and deficiency levels stayed around 60–70% during the whole ICU stay. 25OHD levels did not differ between the patients in both hospitals at any time. The other vitamin D measurements (vitamin D-binding protein (DBP), 25OHD2, 25OHD3, and 24.25diOHD3) can be found in Appendix A.

Vitamin E levels were significantly lower in the COVID-19 patients compared with the control at baseline (*p* = 0.019), but deficiency levels were low. Vitamin E levels increased during the time, and deficiency levels stayed around 0–10% during the whole ICU stay. Vitamin E levels did not differ between the patients in both hospitals at any time.

### 3.2. Trace Elements

Different markers of iron status are shown in Figure 2. At baseline, iron and transferrin concentrations were significantly lower in the COVID-19 patients compared with the controls (*p* < 0.001 for both). In both hospitals, the majority of the patients had decreased concentrations of iron and transferrin at baseline, compared with a minority of the controls (see Appendix A). Ferritin levels were very high in all the COVID-19 patients and the majority of the controls. As ferritin is likely to be elevated as an acute-phase protein, it complicates the diagnosis of iron deficiency in these critically ill patients. As shown in the figure, transferrin saturation (iron concentration divided by TIBC) is mostly <20% in all the COVID-19 patients, which can support the presence of iron deficiency [26] but can also be explained by inflammatory anemia.

Copper, zinc, and selenium levels are shown in Figure 3. Zinc levels were significantly lower in the COVID-19 patients (70–80% deficiency) compared with controls (0% deficiency) at baseline (*p* < 0.001). During ICU stay, zinc levels only significantly increased in the AMC patients. The number of deficient patients decreased to 10–30% in both hospitals. Zinc levels did not differ between the patients in both hospitals at any time. The increase from baseline to week 3 was significantly higher in AMC compared with the increase in the same time period in VUmc (*p* = 0.018 for the interaction term with time).

Selenium levels were significantly lower in the COVID-19 patients (50–60% deficiency) compared with controls (0% deficiency) at baseline (*p* = 0.015). During ICU stay, selenium levels increased, and the number of patients with a selenium deficiency decreased to 0% in both hospitals. Selenium levels did not differ between the patients in both hospitals at any time.

Copper levels were significantly higher in the COVID-19 patients compared with controls at baseline (*p* < 0.001), with 0% deficiency for all patients. Copper levels increased in time, so deficiency stayed 0% during 3 weeks of ICU stay. Copper levels did not differ between the patients in both hospitals at any time.

### 3.3. Degree of Inflammation and Organ Failure during ICU Stay

CRP levels were significantly higher in the COVID-19 patients compared with the controls at baseline (*p* < 0.001; see Figure 4). In both hospitals, mean CRP levels significantly decreased during 3 weeks of ICU stay (from 206 to 133 mg/L in AMC and from 263 to 67 mg/L in VUmc). CRP levels did not differ between the patients in both hospitals at any time. The decrease in CRP from baseline to week 3 was significantly stronger in VUmc compared with the decrease in the same time period in AMC (*p* = 0.032 for the interaction term with time).

There was a negative correlation between the CRP level and levels of vitamin A (r = −0.58, *n* = 85, *p* < 0.001), vitamin E (r = −0.30, *n* = 85, *p* = 0.005), iron (r = −0.60, *n* = 85, *p* < 0.001), transferrin (r = −0.66, *n* = 85, *p* < 0.001), selenium (r = −0.52, *n* = 85, *p* < 0.001), and zinc (r = −0.54, *n* = 85, *p* < 0.001). No correlation was found for vitamin B6 (*p* = 0.094), total 25OHD (vitamin D) (*p* = 0.334), and copper (*p* = 0.866).

In AMC, the SOFA scores significantly decreased from baseline to week 1 (*p* = 0.010), being significantly lower compared with VUmc at week 1 (*p* = 0.035) (Figure 5). No other differences were found between the patients in both hospitals. In VUmc, the SOFA scores did not significantly change during ICU stay.

There was a negative correlation between SOFA scores from the COVID-19 patients and total 25OHD (vitamin D) (r = −0.38, *n*= 77, *p* < 0.001) and selenium (r = −0.41, *n* = 77, *p* < 0.001).

## 4. Discussion

This is the first study of serial measurements of multiple vitamins and trace elements in critically ill COVID-19 patients during three weeks of ICU stay. We demonstrate high deficiency rates (50–100%) of vitamins A, B6, and D; zinc; and selenium at ICU admission, along with low iron status. After three weeks of ICU stay, deficiency rates remained high for vitamins B6 and D, and iron status remained low. An increase in vitamins A and E, zinc, and selenium levels was observed over time, concomitant with a decrease in CRP while patients were recovering from their COVID-19. Administering additional micronutrients did not result in higher micronutrient levels or a faster increase in micronutrient levels over time.

The normal levels of vitamin E and copper, and the low levels of vitamins A, B6, and D; iron; transferrin; zinc; and selenium are in line with previous literature [5,6,7,8,9,10,11,12,13,14,15,16,17,18,19]. The course of the micronutrient levels in the blood might have been influenced by the inflammation state of our patients. Inflammation status was high at admission, after which CRP levels significantly decreased during ICU stay from 200–260 mg/L to 70–130 mg/L on average. All micronutrients except vitamins B6 and D and copper were negatively associated with CRP levels. A previous study analyzing many blood samples showed that plasma concentrations of all micronutrients (vitamins A, B6, and D; zinc; and selenium), except copper and vitamin E, decrease as the inflammatory response increases, possibly due to redistribution, with high inter-patient variations [20]. Also, vitamin E concentrations become less interpretable when CRP > 80 mg/L. Copper concentrations even increase as a result of inflammation, as ceruloplasmin is a positive acute phase reactant [20]. In addition, the ESPEN guidelines describe that indices of iron status are affected by inflammation [21]. Our findings might also point toward a (partly) redistribution effect, as plasma vitamins A and E, zinc, and selenium levels normalized when inflammation concomitantly decreased during ICU stay, regardless of additional administration. However, as mentioned in the Introduction, illness itself also leads to decreased total body micronutrient content. This is illustrated by our control patients who had low CRP levels and low plasma levels of vitamins B6 and D and iron. These same micronutrient levels stayed low in the majority of our COVID-19 patients with the resolution of inflammation state. However, it might be that inflammation was still high enough to cause low levels due to redistribution. This impact of systemic inflammation was nicely shown for 25OHD in an in vivo model [27]. Therefore, the exact (combined) impact of (critical) illness and inflammation on the redistribution and total body content of the different micronutrients remains unrevealed. This complicates the diagnosis of true micronutrient deficiencies.

Regarding the severity of illness, our study showed that vitamin D and selenium levels were negatively associated with the SOFA score, a marker of organ failure. A review and meta-analysis in COVID-19 patients described consistent evidence for the association between low vitamin D levels and the severity of disease and mortality [28,29]. In addition, COVID-19 literature described that low vitamin A, selenium, zinc, and iron levels were associated with ARDS and mortality [30], worse outcomes [31], disease severity [16,32], and mortality [16], respectively. The existence of a possible (reverse) causality remains unknown. We did not investigate the relation between micronutrient administration and the (long-term) clinical outcome of COVID-19 patients in our small study group. And up to now, no consistent benefit of most micronutrient administration, solely or combined, has been found in the literature [21,33,34,35,36]. However, a recent meta-analysis showed a protective effect of vitamin D supplementation on ICU hospitalization and possibly on ICU mortality, but this needs further study [37]. A large trial investigating selenium supplementation in COVID-19 patients is currently ongoing [38].

### 4.1. Repletion Studies

In our study, we did not find clinically relevant differences in micronutrient levels between patients receiving enteral nutrition and patients receiving enteral nutrition with additional (intravenous) micronutrients. A previous study in COVID-19 patients showed that oral supplementation of vitamins A, D, and E improved plasma concentrations [33]. However, the doses used were much larger (25,000 IUs vitamin A daily, 600,000 IUs vitamin D once, and 600 IUs vitamin E daily) than the daily additional doses administered in our study. No iron and copper repletion studies have been performed in COVID-19 patients. Another trial showed that administering 0.24 mg/kg zinc per day for seven days increased zinc levels above the deficiency cut-off compared with placebo [17]. In addition, intravenously administering daily 1 mg selenite next to fortified nutrition containing zinc and selenium improved zinc and selenium levels after 10 to 14 days on the ICU in COVID-19 patients [18]. Unfortunately, no control patients were enrolled, so a spontaneous restoration of zinc and selenium levels cannot be ruled out. An association between selenium levels and CRP was also found in that study. However, the fact that other studies were able to find a supplementation effect with higher doses could indicate that our administered doses were too low or the duration of the intravenous administration of five days was too short.

### 4.2. Strengths and Limitations

This is the first study that measured micronutrient plasma concentrations during a prolonged time of three weeks of ICU stay. Furthermore, we explored the effect of additional (intravenous) micronutrient administration during three weeks of ICU stay. The lack of difference between the groups might be caused by the small groups and small differences in received doses. In addition, the differences in administered doses between the described trial that did find an effect [33] and our study were large. Unfortunately, the sample size of this pilot study was too small to include CRP and the SOFA score in the mixed model analysis. With mixed model analysis, the impact of CRP and the SOFA score on the course of different micronutrient plasma levels during a long ICU stay can be more accurately assessed.

### 4.3. Future Directions

It would be of high interest to measure micronutrient levels in other matrices, e.g., in cells, to gather more information about the true total body micronutrient content. This can give insights in the exact micronutrient requirements of adult ICU patients and can create a more personalized approach in high-risk patient groups. This opposes a blind one-size-fits-all supplementation strategy, where one should be very careful because there is a risk of toxicity due to over-supplementation in case of apparent deficiencies.

## 5. Conclusions

This pilot study in COVID-19 ICU patients shows high deficiency rates of vitamins A, B6, and D; zinc; and selenium at ICU admission, along with low iron status. Levels of vitamins A and E, zinc, and selenium improved after three weeks of ICU stay, except for vitamins B6 and D and iron. Administering additional micronutrients did not result in higher micronutrient levels or a faster increase in micronutrient levels over time. The observed spontaneous increase in micronutrient levels was associated with the resolution of inflammation state over time, suggesting that low levels, at least partly, can be explained by redistribution rather than true deficiencies.

## Figures and Tables

**Figure 1 nutrients-16-00385-f001:**
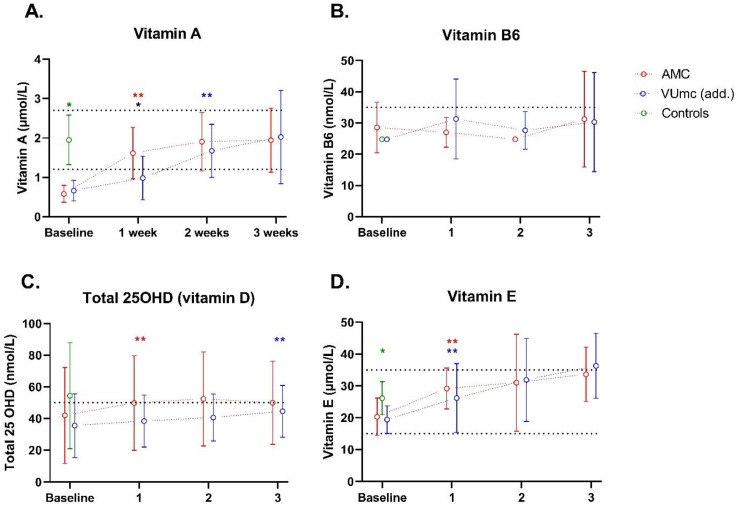
Vitamin status of 10 patients in AMC, 10 patients in VUmc (add.), and 5 control patients during ICU stay. The dashed line(s) indicate the (upper and) lower limit of the normal levels. Data are presented as mean ± standard deviation. * indicates a significant difference between the patient group (*n* = 20) and control group (*n* = 5); * indicates a significant difference between the two hospitals (*n* = 10 vs. *n* = 10); ** or ** indicates a significant difference within the same hospital between baseline and the indicated time point (*n* = 10 vs. *n* = 10). (**A**) Vitamin A; (**B**) Vitamin B6; (**C**) Total 25OHD (vitamin D); (**D**) Vitamin E.

**Figure 2 nutrients-16-00385-f002:**
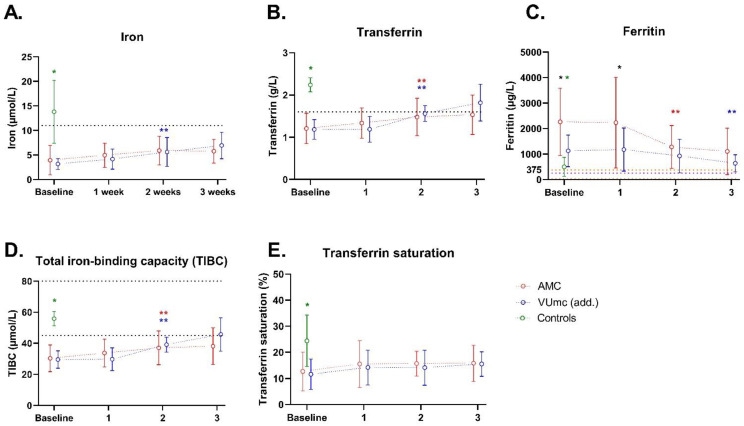
Iron status of 10 patients in AMC, 10 patients in VUmc (add.), and 5 control patients during ICU stay. The dashed line(s) indicate the (upper and) lower limit of the normal levels. In (**C**), the yellow dashed lines indicate the normal window for women, and purple dashed line for men. Data are presented as mean ± standard deviation. * indicates a significant difference between the patient group (*n* = 20) and control group (*n* = 5); * indicates a significant difference between the two hospitals (*n* = 10 vs. *n* = 10); ** or ** indicates a significant difference within the same hospital between baseline and the indicated time point (*n* = 10 vs. *n =* 10). (**A**) Iron; (**B**) Transferrin; (**C**) Ferritin; (**D**) TIBC; (**E**) Transferrin saturation.

**Figure 3 nutrients-16-00385-f003:**
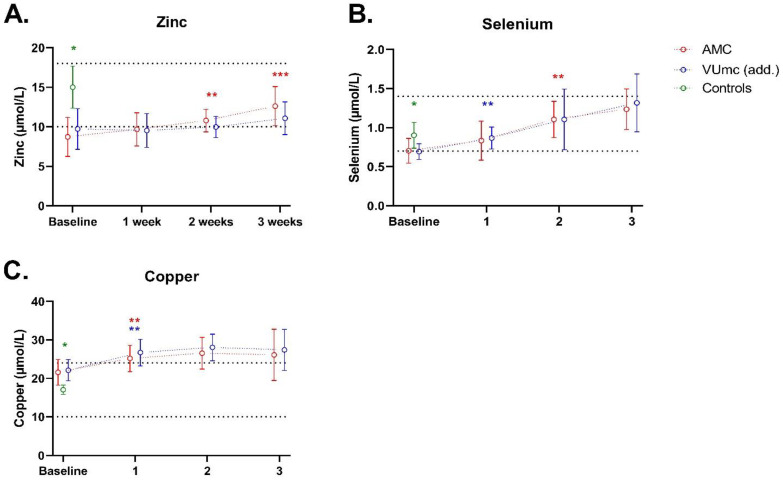
Zinc, selenium, and copper status of 10 patients in AMC, 10 patients in VUmc (add.), and 5 control patients during ICU stay. The dashed lines indicate the upper and lower limits of the normal levels. Data are presented as mean ± standard deviation. * indicates a significant difference between the patient group (*n* = 20) and control group (*n* = 5); ** or ** indicates a significant difference within the same hospital between baseline and the indicated time point (*n* = 10 vs. *n* = 10); *** indicates a significant difference in the increase or decrease over time between the hospitals. This was only evaluated between baseline and week 3. (**A**) Zinc; (**B**) Selenium; (**C**) Copper.

**Figure 4 nutrients-16-00385-f004:**
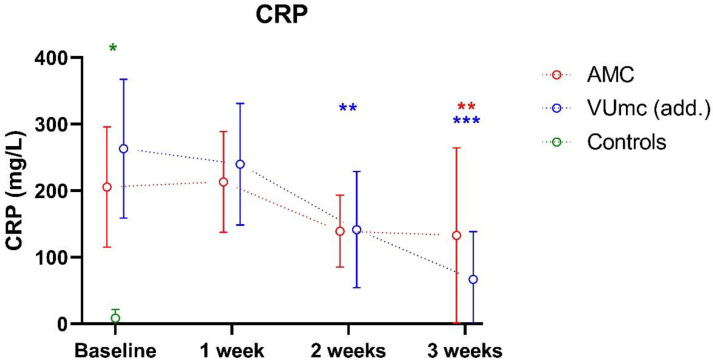
CRP level, as marker of inflammation, of 10 patients in AMC, 10 patients in VUmc (add.), and 5 control patients during ICU stay. Data are presented as mean ± standard deviation. * indicates a significant difference between the patient group (*n* = 20) and control group (*n* = 5); ** or ** indicates a significant difference within the same hospital between baseline and the indicated time point **, and week 1 and the indicated time point ** (*n* = 10 vs. *n* = 10). *** indicates a significant difference in the increase or decrease over time between the hospitals. This was only evaluated between baseline and week 3.

**Figure 5 nutrients-16-00385-f005:**
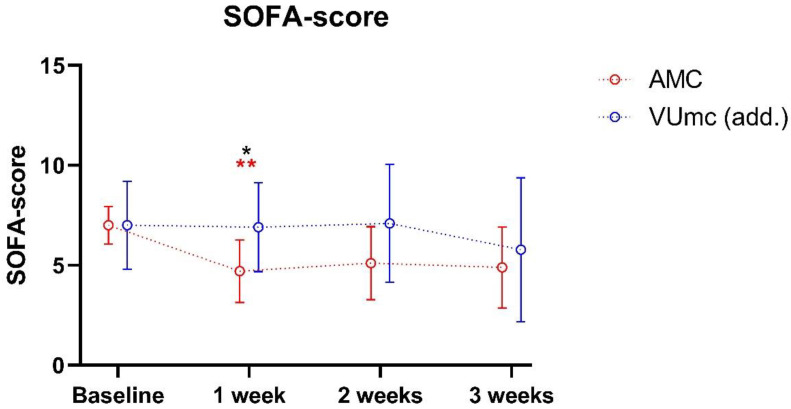
SOFA scores of both patient groups during ICU stay. Data are presented as mean ± standard deviation. ***** indicates a significant difference between the two hospitals (*n* = 10 vs. *n* = 10); ** indicates a significant difference within the same hospital between baseline and the indicated time point (*n* = 10 vs. *n* = 10).

**Table 1 nutrients-16-00385-t001:** Baseline characteristics.

	Patients AMC (*n =* 10)	Patients VUmc (*n =* 10)	Controls AMC (*n =* 5)
Age (years)	61 [58–71]	66 [59–75]	31 [24–78]
Sex, men (%)	7 (70)	8 (80)	3 (60)
BMI (kg/m^2^)	26.9 [21.0–29.0]	28.6 [26.1–30.9]	24.8 ^b^
**Medical history**			
Chronic cardiac disease	3	3	1
Hypertension	6	3	0
Chronic pulmonary disease	3	0	0
Asthma	2	2	0
Immunosuppressive medication	1	1	0
Obesity	1	4	n.a.
Diabetes without complications	4	2	0
Rheumatologic disorder	1	0	0
Autoimmune and/or inflammatory diseases	0	1	0
**Hospital admission**			
APACHE II	17 [16–19]	14 [10–18]	10 [0–17] ^c^
APACHE IV	62 [45–78]	62 [52–68]	42 [13–68] ^c^
SOFA baseline ^a^	7 [6–8]	7 [7–8]	2 [0–4] ^c^
Hb baseline	8.3 [7.3–9.0]	7.4 [6.2–8.4]	8.9 [8.3–9.1]
Creatinine baseline	92 [70–118]	73 [65–106]	61 [55–72]
eGFR baseline	80 [47–90]	88 [64–90]	>60
CRP baseline (mg/L)	209 [129–282]	255 [200–352]	2 [1–20]
Albumin baseline (g/L)	22.0 [19.6–27.1]	20.6 [18.7–25.0]	39.6 [37.5–40.7]
Mechanical ventilation (%)	100%	100%	40%
Vasopressor support (%)	100%	100%	0%
Total length of stay ICU (days)	30 [23–46]	27 [24–36]	1 [0–2]
Dexamethasone treatment (%)	0%	20%	0%
Tocilizumab treatment (%)	0%	0%	0%

APACHE: Acute physiology and chronic health evaluation; BMI: body mass index; n.a.: not available; SOFA: sequential organ failure assessment. Data are presented as mean ± standard deviation or as median with [interquartile range]. ^a^ SOFA scores are calculated without the central nervous system score due to its unreliability when patients receive sedatives. SOFA scores were calculated according to the NICE criteria [25]. ^b^ Only available for one patient. ^c^ Many components of the ICU scores were unavailable for these control patients, admitted to the ER. The SOFA score was only determined for those control patients who were admitted to the ICU afterward.

## Data Availability

The data are available from the corresponding author on reasonable request. The data are not publicly available due to the extensive size of the datasets. Therefore, we prefer to deliver specific data for specific requests.

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
