# Peer review of "Micronutrient Status of Critically Ill Patients with COVID-19 Pneumonia"

_nutrients, 2024, doi:10.3390/nu16030385_

Round 1
Reviewer 1 Report
Comments and Suggestions for Authors
In a study performed in the Netherlands, Rozemeijer et al. evaluated the micronutrient status (vitamin A, B6, D, E, iron, zinc, copper, and selenium) of critically ill patients with COVID-19 pneumonia. Despite having evaluated a limited sample size (n=20 patients and n=5 controls), the study is interesting and clinically relevant, but I have some suggestions for improvements of the manuscript:
- Introduction: Describe more information about the current knowledge regarding nutritional deficiencies and severe COVID-19.
- Methods, Study population and design: Describe more details about the team involved in the patients’ nutritional care. Was it performed only by medical doctors or did patients receive care from clinical nutritionists? Multidisciplinary treatment involving registered dietitians/nutritionists is essential in these cases. Make this evident in the article.
- Results, line 152: "In total, 25 patients were included". Here the word "patients" can confuse the reader. I suggest writing something like, "In total, 20 COVID-19 patients and 5 controls were included".
- Results, Table 1 and text lines 165-166: The mean age of patients and controls is quite different. This is an important limitation that needs to be highlighted in the study. Age is not comparable between the groups. Please, revise it in the text.
- Results, Figure 4: CRP levels of controls are not shown in the figure. Please revise or explain it.
- Discussion, lines 282-283: "Administering additional micronutrients did not result in higher micronutrient levels or a faster increase in micronutrient levels over time". These findings are unexpected, intriguing and should be discussed further based on the literature.
Comments on the Quality of English LanguageEnglish is fine for scientific purposes, but minor corrections are needed.
Reviewer 2 Report
Comments and Suggestions for Authors
As mentioned in the background, you have presented thought-provoking results on trace elements that are difficult to evaluate in real time. I consider it a useful report.
I would ask for additional information on this subject.
Enteral nutrition were administered to patients admitted to the ICU, is there a protocol for administration at each the two hospitals?
If all patients were on protocol for the duration of your ICU stay, is it possible to find out the enteral nutrition dosage for all patients?
I think this is important information because the enteral feedings used contain a reasonable amount of micronutrients.
All patients with severe COVID-19 have been in the ICU for more than 3 weeks, but were they on enteral nutrition for all of that time? Or were those patients whose condition had improved taking it orally?
What is the median duration of enteral feeding?
(1)  Is it possible to determine the median dietary intake of patients who are able to take food orally?
(2)  Could you present with additional research what the sufficiency of micronutrients doses from enteral nutrition + supplementation compared to the recommended doses for adults in your country?
I think this is a factor that may influence this study. However, it is quite difficult to investigate (1) and (2). I suggest that this point could be mentioned as limitations.
Patient outcomes in this report What is the mortality rate at 30 days or at the end of the study?
Round 2
Reviewer 2 Report
Comments and Suggestions for Authors
I have read the revised version.
Thank you also for your thoughtful reply to my comments.
Add the statement "in accordance with each hospital's protocol" to lines 90-93 of the method,
Could you present that protocol in the supplement data?
If you could show us that protocol, I think it would give us a rough idea, even if we don't know the actual detailed amounts of nutritional supplements administered.
Author Response
Response
A point-by-point response on the reviewer’s comments is provided below.
Reviewer 2:
Comments
I have read the revised version.
Thank you also for your thoughtful reply to my comments.
Add the statement "in accordance with each hospital's protocol" to lines 90-93 of the method,
We added this statement.
Could you present that protocol in the supplement data?
If you could show us that protocol, I think it would give us a rough idea, even if we don't know the actual detailed amounts of nutritional supplements administered.
Our protocols have been written only in Dutch and, as it will still be not 100% sure how the protocol was followed, we decided to make an additional supplementary file in which we present the (enteral) nutrition each patient received approximately. This will give the reader the most accurate idea about the micronutrients provided. In line 90-91 and line 104-105 additional information about this is provided:
‘See Supplementary File 2 for an overview of the standard enteral nutrition each patient received in each hospital.’